# Melanocortin 1 Receptor (MC1R): Pharmacological and Therapeutic Aspects

**DOI:** 10.3390/ijms241512152

**Published:** 2023-07-29

**Authors:** Yoonwoo Mun, Woohyun Kim, Dongyun Shin

**Affiliations:** 1College of Pharmacy, Gachon University, 191 Hambakmoe-ro, Yeonsu-gu, Incheon 21936, Republic of Korea; ansqosdl@naver.com (Y.M.); kimwoohyun132@gmail.com (W.K.); 2Gachon Pharmaceutical Research Institute, Gachon University, 191 Hambakmoe-ro, Yeonsu-gu, Incheon 21936, Republic of Korea

**Keywords:** melanocortin, receptor, peptide, modulator, disease

## Abstract

Melanocortins play crucial roles in regulating the stress response, inflammation, and skin pigmentation. In this review, we focus on the melanocortin 1 receptor (MC1R), a G protein-coupled receptor primarily known for regulating skin pigmentation and exhibiting anti-inflammatory effects. First, we provide an overview of the structure, signaling pathways, and related diseases of MC1R. Next, we discuss the potential therapeutic use of synthetic peptides and small molecule modulators of MC1R, highlighting the development of various drugs that enhance stability through amino acid sequence modifications and small molecule drugs to overcome limitations associated with peptide characteristics. Notably, MC1R-targeted drugs have applications beyond skin pigmentation-related diseases, which predominantly affect MC1R in melanocytes. These drugs can also be useful in treating inflammatory diseases with MC1R expression present in various cells. Our review underscores the potential of MC1R-targeted drugs to treat a wide range of diseases and encourages further research in this area.

## 1. Introduction

Melanocortins are a group of hormones that play crucial roles in regulating various physiological processes, including stress response, inflammation, and skin pigmentation. These hormones are polypeptides derived from pro-opiomelanocortin (POMC) and include adrenocorticotropin hormone (ACTH), α-, β-, and γ-melanocyte stimulating hormone (α-, β-, and γ-MSH) [1,2]. While the functions of ACTH [3] and α-MSH are well-established in skin pigmentation [4], anti-inflammatory [5], and microbicidal characteristics [6], the role of β-MSH and γ-MSH are less well understood. Some studies suggested that they possess anti-inflammatory properties [7,8,9,10].

The functions of melanocortin peptides are mediated by G protein-coupled receptors (GPCRs), primarily the stimulatory G protein (Gs) [11]. The melanocortin receptor family is the smallest member of class A, a rhodopsin-like family of GPCRs, and consists of five isotypes (MC1R, MC2R, MC3R, MC4R, and MC5R) with varied tissue expression and functions [12]. MC1R, mainly expressed in both melanocytes and leukocytes, enhances UV resistance and anti-inflammatory signaling when activated. MC2R is found in the adrenal cortex, and both MC3R and MC4R, which are reported to control food intake and sexual function, are largely present in the CNS. MC5R, located in the brain and skeletal muscle, has an exocrine role. The low sequence homology (40–60%) among the five receptors explains the lack of ligand selectivity [13,14,15,16].

The MC1R is a well-known receptor for α-MSH expressed in the skin and hair follicles, where it controls pigmentation. However, recent studies have revealed that MC1R is also expressed in various other cell types that are susceptible to the anti-inflammatory effects of melanocortins [17,18]. Thus, understanding the regulation of MC1R could potentially lead to the development of novel therapeutic strategies. This review article provides a brief overview of the structure, signaling pathway, and related diseases associated with MC1R and discusses the potential of synthetic peptide modulators and small molecule modulators of MC1R as therapeutic agents. Through a detailed examination of MC1R and its modulators, this review aims to provide insights into the potential clinical implications of modulating MC1R activity.

## 2. Melanocortin 1 Receptor (MC1R)

### 2.1. Structure

The human MC1R is primarily found on melanocytes and malignant melanoma cells and consists of 317 amino acids [19,20,21]. While the normal expression of MC1R protein is low, melanocytes express approximately 700 protein units, with slightly higher amounts found on melanoma cells [22,23]. MC1R is a GPCR with seven α-helical transmembrane (TM) domains, an N-linked glycosylation site at the external N-terminus, a palmitoylation site at the intracellular C-terminus, and a DRY motif at the junction of the third TM domain. Unlike other GPCRs, the first and second extracellular domains of the MC receptor subfamily lack one or two cysteines, while the fourth and fifth TM domains lack proline. The intracellular and transmembrane domains of MC1R regulate adenylyl cyclase connections and signaling, while the extracellular and transmembrane domains interact with MC1R ligands.

The extracellular N-terminal tail serves as a signal anchor and plays a crucial role in ligand affinity. A conserved cysteine residue at the intersection of the N-terminus and the first TM domain is crucial for receptor function [24,25,26,27,28]. The C-terminus is involved in receptor interactions with the G protein at the plasma membrane, as well as protein trafficking from the endoplasmic reticulum to the plasma membrane. Also, C-terminus affects desensitization, internalization, and the plasma membrane location of the receptor [29,30,31,32,33].

The intracellular and extracellular loops (ils and els, respectively) lie between the transmembrane domains of the melanocortin 1 receptor. These loops share conserved sequences with other MC receptors. Despite being smaller than most GPCRs, MC1R els are essential for constitutive basal signaling activity. Mutations in this area affect binding affinity, as the els of the MC1R interact with ligands. Due to conserved proline and cysteine residues, els appear to be essential for melanocortin affinity. Similarly, MC1R ils play a crucial role in Gs protein binding and have phosphorylation sites that impact signal modulation, internalization, and receptor cycling [25,27,28,31,34].

Recently, the Cryo-electron microscopy (Cryo-EM) structure of MC1R and the MC1R–Gs complexes bound to the endogenous hormone α-MSH, the marketed drug afamelanotide, and a synthetic agonist called SHU9119 were determined [35]. These findings may pave the way for resolving the lack of selectivity in MC drug discovery.

### 2.2. Signaling Pathway

The melanocortin 1 receptor (MC1R) is a receptor that forms a complex with heterotrimeric G proteins. When agonistic ligands bind to MC1R, the Gαs protein is separated, and MC1R activates adenylyl cyclase, which leads to the production of cAMP, a crucial second messenger that regulates many cellular processes. In melanocytes, cAMP activates protein kinase A (PKA) and triggers downstream signaling pathways that activate different effector pathways, including the CREB and MITF networks. These pathways lead to the increased expression of tyrosinase and dopachrome tautomerase, two enzymes that are involved in melanin synthesis, resulting in the production of melanin. The melanin produced is then transmitted to nearby keratinocytes, creating a protective layer that improves the skin’s ability to prevent further UV damage. Moreover, the increase in cAMP levels in melanocytes enhances antioxidant defenses and accelerates nucleotide excision repair (NER), which is vital for safeguarding the skin from UV damage (Figure 1) [36,37,38,39,40,41,42,43].

Moreover, it is also reported that there is a cAMP-independent pathway mediated by cKIT, which is not discussed in this review. This pathway activates the extracellular signal-regulated protein kinases 1 and 2 (ERK 1/2) by triggering the NRAS-BRAF-MEK-ERK cascade and active ERKs can phosphorylate MITF. Furthermore, it has been suggested that the activation of AKT downstream of α-MSH may also occur via cKIT [44].

Furthermore, the MC1R signaling axis has an anti-inflammatory effect through downstream pathways. When α-MSH binds to MC1R, it triggers the production of intracellular cAMP, which activates protein kinases C and A, leading to the activation of the MAPK and JAK-STAT pathways. These pathways prevent IκB degradation and activate CREB, a transcription factor that regulates anti-inflammatory mediators such as IB and IL-10. In addition, protein kinase activation enhances the levels of cytoplasmic IκB, inhibiting the expression of downstream pro-inflammatory genes, including IL-1, TNF-a, IL-6, IL-8, and IL-12, iNOS, and adhesion molecules (ICAM-1, VCAM-1, and MMPs) (Figure 2) [45,46,47,48,49,50,51,52,53,54,55,56,57,58,59,60].

## 3. MC1R-Related Diseases

### 3.1. Melanoma

Melanoma is a highly aggressive and resistant cancer that results from the malignant progression of melanocytes, which are pigment cells located in the skin’s epidermis [61]. The traditional chemotherapy treatments have demonstrated limited efficacy in treating melanoma, which highlights the need for innovative therapeutic approaches that can take advantage of the distinctive features of these tumors [62]. One such characteristic is the overexpression of the MC1R cell surface endocytic receptor on the surface of human melanomas, which makes it a crucial tumor marker [63,64]. The activation of MC1R is necessary to regulate melanocyte cell division and melanin production, which provides the skin with protection from UV radiation. Additionally, MC1R has potential applications in diagnostics and targeted drug delivery [65,66]. Innovative treatment approaches are currently being developed based on MC1R for melanoma. MC1R expression is frequently elevated in both melanoma cell lines and melanomas, and contrary to earlier claims that MC1R activation could cause melanoma, recent scientific evidence demonstrates that this receptor not only does not cause melanoma but that its activation can actually stimulate DNA repair processes that have the potential to prevent the disease [67,68,69]. In addition, the MC1R is also implicated in vitiligo, a pigmentation disorder characterized by the loss or destruction of melanocytes in the skin and hair follicles. This results in the inability to produce melanin, which is necessary for normal pigmentation. Thus, MC1R emerges as a potential therapeutic target for vitiligo since MC1R plays a crucial role in regulating skin pigmentation. Recently, a novel MC1R peptide agonist is reported, and we will introduce it in the synthetic peptide modulators section [70].

### 3.2. Systemic Sclerosis

Systemic sclerosis (SSc), or scleroderma, is an autoimmune disorder characterized by dysregulation of the immune system and inflammation, microvascular dysfunction, and widespread fibrosis in various organs [71]. While no approved medications exist for SSc apart from those for SSc-associated ILD (interstitial lung disease), there is a pressing need for innovative treatments that can effectively target fibrosis in multiple organs, including the skin [72,73,74]. Activating the melanocortin 1 receptor has been shown to have broad anti-inflammatory and antifibrotic effects. Thus, a recent study investigated the potential of a novel oral MC1R agonist, dersimelagon (MT-7117), as a treatment for SSc. In preclinical SSc models, MT-7117 exhibited disease-modifying effects. Target expression analysis and research into its mode of action indicated that MT-7117 has a favorable effect on inflammation, vascular dysfunction, and fibrosis, all of which are key pathologies in SSc. These findings suggest that MT-7117 may have potential as a treatment for SSc. A phase 2 clinical trial is currently underway to evaluate the efficacy and tolerability of MT-7117 in patients with early, progressive diffuse cutaneous SSc [74]. More information about dersimelagon will be discussed in the section on small molecule modulators.

### 3.3. Neuroinflammation

Neuroinflammation is a critical factor in the progression of neurological damage caused by hypoxic-ischemic (HI) events, and microglia play a significant role in this process [75,76]. In various neurological disorders, MC1R activation has been shown to have anti-inflammatory and neuroprotective effects [77,78]. Recently, researchers investigated the potential of MC1R activation to mitigate neuroinflammation and repair neurological impairments in neonatal rats with hypoxic-ischemic neurological damage [79]. The results showed that BMS-470539, an MC1R activator, reduced neuroinflammation and repaired neurological impairments in these rats. The study also revealed that the anti-inflammatory and neuroprotective effects of MC1R activation were mediated through the MC1R/cAMP/PKA/Nurr1 signaling pathway. The findings suggest that MC1R activation could be a promising therapeutic target for the treatment of hypoxic-ischemic encephalopathy (HIE) in newborns. Additional details regarding BMS-470539 will be provided in the small molecule modulators section.

### 3.4. Atherosclerosis

Monocytes and macrophages are known to express the MC1R, which mediates anti-inflammatory effects and helps prevent macrophage foam cell production by increasing cholesterol efflux via ABCA1 and ABCG1 transporters [80,81,82]. Recently, a study investigated whether a systemic deficiency of MC1R signaling affects the development of atherosclerosis. The study highlighted the significant role of MC1R in the development of atherosclerosis. The findings suggested that a lack of MC1R signaling may exacerbate atherosclerosis by disrupting cholesterol transport and increasing arterial monocyte deposition [83]. Furthermore, MC1R plays a role in immunomodulation. Specifically, α-MSH has been shown to decrease pro-inflammatory cytokines in various pulmonary inflammatory disorders, including asthma, sarcoidosis, and acute respiratory distress syndrome. Animal models of pulmonary fibrosis have also demonstrated that α-MSH can reduce fibrogenesis [84]. In addition to the diseases previously mentioned, MC1R has also been implicated in other diseases, such as intestinal and ocular inflammation [85] and Parkinson’s disease [86]. Recent research has shown that modulating MC1R may have therapeutic potential for the diseases previously mentioned.

## 4. Modulators

### 4.1. Endogenous Ligands

After UV damage, melanocortins such as α-MSH and ACTH are induced in the skin, and these ligands protect the skin by binding to MC1R and triggering changes in melanocytes that enhance their resistance to UV (Figure 3) [87,88,89]. However, the MC1R can also be affected by other ligands, such as agouti signaling protein (ASIP) and β-defensin 3 (βD3), which can have a significant impact on MC1R signaling [90,91,92,93]. Melanocortins stimulate MC1R signaling, while ASIP inhibits MC1R signaling and reduces cAMP levels [94,95]. In contrast, βD3 has little effect on signaling but can act as a competitive inhibitor, interfering with the binding of α-MSH or ASIP [96,97]. Synthetic modulators of MC1R, including both peptides and small molecules, have been reported, and now some of these modulators will be introduced.

### 4.2. Synthetic Peptide Modulators

As previously explained, the endogenous ligands for melanocortin receptors are ACTH and α-, β-, γ-MSH. The most important finding in early drug development was that the amino acid sequence, His6-Phe-Arg-Trp9, was the key sequence needed to activate the melanocortin receptor [98]. And the sequence was modified to solve the instability, which was a problem of α-MSH, leading to the development of [Nle4, D-Phe7]-α-MSH (melanotan I) [99,100]. Additionally, studies on the cyclization of peptides were conducted to obtain melanotan II using NDP-MSH as a scaffold [101,102]. With these findings, efforts are currently underway to produce selective and stable peptides on the melanocortin 1 receptor, such as PL-8177 [67]. Among them, we would like to introduce melanotan I and II, which are key substances in the development of peptide drugs (Figure 4).

#### 4.2.1. Melanotan I (MT-I)

Melanotan I is an early analog of α-MSH that acts as a non-selective agonist of melanocortin receptors and stimulates melanin production. During the development of peptide-based drugs, researchers focused on the common sequence of MSH, namely His6-Phe-Arg-Trp9, and further studies were conducted on this sequence (Figure 4) [98]. Eventually, in the frog skin bioassay, it was observed that Ac(Acetyl)-α-MSH7-10-NH2 and Ac-α-MSH6-8-NH2 had no activity, while Ac-α-MSH6-9-NH2 did. This led to the discovery that the minimal sequence required for the biological activity of α-MSH is His6-Phe-Arg-Trp9 [99]. Furthermore, to increase the stability of the initial α-MSH analog, Sawyer et al. replaced Phe at position 7 with D-Phe and Methionine at position 4 with the amino acid norleucine to prevent oxidation of methionine, as highlighted in green ball. These changes resulted in the production of [Nle4, D-Phe7]-α-MSH (NDP-MSH, melanotan I), a peptide with higher potency and stability than α-MSH [100].

When melanotan I activates MC1R, cAMP is produced, and it activates microphthalmia transcription factor (MITF) expression, which induces the expression of enzymes for eumelanin production. This process increases the production of eumelanin in melanocytes. In addition, melanotan I activates tyrosinase and induces an increase in eumelanin content in melanocytes [103,104,105].

Afamelanotide, well-known as the international nonproprietary name of melanotan I, has been used in patients with erythropoietic protoporphyria (EPP) since 2019. EPP is a disease that causes abnormal hemoglobin synthesis in red blood cells, which can cause skin damage even with a little sunlight. When afamelanotide binds to MC1R, it activates melanocytes and stimulates eumelanin synthesis. Eumelanin protects against UV light by absorbing and scattering it, scavenging free radicals and reactive oxygen species, and acting as a neutral density filter capable of decreasing transmission of all wavelengths of light [106,107]. Recent studies have shed light on the binding structure of MC1R and ligands, while no information has yet been disclosed about the binding structure of receptors and peptides. Afamelanotide, the compound that was switched from Phe7 to D-Phe7 in α-MSH, has an extra hydrogen bond with the TM2 domain in D-Phe7. It has been demonstrated that afamelanotide has a higher affinity for receptors than α-MSH, as evidenced by its capacity to make cAMP, which is superior to α-MSH, as determined by various mutants (Figure 5) [35].

#### 4.2.2. Melanotan II (MT-II)

Melanotan II is a cyclic peptide that is derived from melanotan I (Figure 6). In particular, the peptide is modified by replacing Glu5 with Asp and Gly10 with Lys. Previous studies by Obeidi and Hadley have demonstrated that cyclic peptides are more stable and potent than linear peptides. Their research investigated the efficacy and stability of a linear form of Ac-[Nle4, D-Phe7, Lys10]-α-MSH4-10-NH2 and a cyclic form, in which Asp5 and Lys10 were linked. The findings of their bioassay demonstrated that cyclic peptides exhibited greater potency and stability than linear peptides [101,102]. Melanotan II has been shown to activate melanocytes as an MC1R agonist, but it is weaker than melanotan I. However, melanotan II was proven to have a stronger sexual effect [108]. It has been reported to induce penile erections in male rats and sexual inspiration [109,110] and increases the sexual activities of female rats [111]. Bremelanotide (PT-141), an acid derivative of melanotan II, acts as an agonist of the MC1R and MC4R and has been demonstrated to increase erectile function and sexual desire (Figure 6) [112,113].

In 2020, Zhou et al. reported an attractive study in which melanotan II was utilized as a probe for selective drug delivery [114,115]. The authors synthesized and designed ligand-drug conjugates with the melanocortin 1 receptor (MC1R) agonist melanotan-II (MT-II) to couple a cytotoxic drug, camptothecin with low cancer resistance (Figure 7). The drug-MT-II conjugates efficiently bound to MC1R and selectively delivered drugs to A375 melanoma cells in vitro. Camptothecin-MT-II was used to study the inhibitory activity on A375 melanoma cells and yielded an IC50 of 16 nM. This approach of drug-MT-II conjugates offers more options in cytotoxic drug selection and is noteworthy for potentially overcoming the cancer-resistant problem of melanoma.

#### 4.2.3. Peptide Agonist for Vitiligo Treatment

As a key regulator of skin pigmentation, MC1R may be an effective therapeutic target for vitiligo. However, α-MSH not only binds to MC1R but also other melanocortin receptors. This leads to diverse physiological changes including sexual function and immunomodulation. Furthermore, the short half-life of peptide drugs has been suggested as a major obstacle. This emphasizes the need for potent and highly selective MC1R agonists, distinct from α-MSH, to effectively treat vitiligo.

In 2023, Zhu et al. introduced a novel MC1R agonist capable of self-assembling into nanofibrils (Figure 8). This peptide, known as Peptide 1, forms a hydrogel that exhibits enhanced stability compared to free peptides. Notably, this hydrogel demonstrates resistance to enzymatic proteolysis. In vitro and ex vivo experiments have convincingly shown that Peptide 1 stimulates MC1R and enhances melanin production by activating tyrosinase and tyrosinase-related protein. Researchers propose that these findings support the MC1R agonist hydrogel as a promising treatment for vitiligo. Moreover, the peptide hydrogel shows potential for preventing UV-induced melanoma and other skin cancers. Consequently, Zhu et al. suggest that this selective and potent MC1R agonist hydrogel presents an alternative for the treatment or prevention of skin pigmentation disorders such as EPP and melanoma [70].

### 4.3. Small Molecule Modulators

During the development of peptide-based drugs, challenges such as low stability of enzymes and selectivity for receptors were encountered. For example, the peptide agent, melanotan I (afamelanotide) is a non-selective agonist of melanocortin receptors except for MC2R [116]. To address the limitations of peptide-based therapeutics, small molecule development has been rapidly evolving, utilizing peptides as starting scaffolds and screening compound libraries [117]. As a result, BMS-470539, derived from an MC4R agonist, has been developed [118]. Additionally, various agonists, including AP1189 and CD08108, are under development [119,120]. Recently, dersimelagon (MT-7117), which selectively acts on MC1R, has been approved in Europe for the treatment of EPP and also has an anti-inflammatory effect [74,116]. Furthermore, JNJ-10229570, an antagonist of the melanocortin 1, 5 receptors, has been developed, and it has been reported to affect sebaceous lipid production [121].

#### 4.3.1. BMS-470539

BMS-470539 is a small molecule agonist selective for MC1R, which was synthesized by modifying the structure of a known MC4R agonist (Figure 9). In a study conducted by Timothy et al., the selectivity for receptors of several compounds was measured by comparing their activity using EC50 values of other isotypes. The EC_50_ (nM) value for MC1R of Compound A (BMS-470539) was 28 ± 12 nM, which was higher than Compound B, Compound C, and Compound D, indicating relatively low activity compared to other compounds (Table 1). However, when expanded to other receptors, it was found that Compound **1** did not activate MC3R, and the selectivity for MC4R and MC5R was significantly lower than that of MC1R. Also, the other compounds had receptors with relatively similar efficacy to MC1R and had relatively high intrinsic activity (IA) for all receptors. This suggests that other compounds may have higher activity but lower selectivity than compound **1** [118]. Therefore, Compound **1**, known as BMS-470539, was identified as having much better selectivity than other compounds (Table 1).

BMS-470539 is also known for its anti-inflammatory effects. BMS-470539 was administered to mice in which an immune response was induced by LPS, and the amount of TNF-α, an inflammatory factor, was measured. The results showed a rapid decrease of up to 92 percent at 100 μmol/kg of BMS-470539 [118]. Kang et al. investigated the activity of NF-κB associated with inflammation to prove the anti-inflammatory effect of BMS-470539 and found a noticeable decrease in the activity of NF-κB. Moreover, it induced a decrease in the infiltration of white blood cells in the lungs of mice by LPS [78]. Recently, BMS-470539 was found to decrease the phosphorylation of p38, ERK1/2, and JNK involved in the MAPK pathway associated with inflammation in neutrophils stimulated by LPS [122].

#### 4.3.2. MT-7117 (Dersimelagon)

MT-7117, or dersimelagon, is a selective agonist of MC1R that has been found to have a much higher selectivity for MC1R than NDP-MSH (afamelanotide), a peptide effective in treating erythropoietic protoporphyria (EPP). In binding affinity and agonistic activity measurement experiments conducted by Suzuki et al., MT-7117 was demonstrated to have a Ki value of 2.26 nM for hMC1R (human MC1R), with much higher selectivity ranging from at least 15 to as much as 700 times compared to other receptors (Figure 10). On the other hand, NDP-MSH showed only a 10-fold difference in selectivity for MC1R compared to other receptors, which suggested that NDP-MSH has lower selectivity than MT-7117 [123]. Due to issues with enzyme stability associated with peptide drugs and higher selectivity for MC1R, MT-7117 has been recently approved in Europe as an alternative treatment for EPP [116].

A recent study revealed that MT-7117 exhibits anti-inflammatory effects [74]. The study showed that daily oral administration of MT-7117 to rats with fibrosis induced by bleomycin significantly delayed skin fibrosis and lung inflammation. In addition, it was found to significantly reduce the expression of ACTA2 (α-smooth muscle actin) mRNA in human skin fibroblasts that had been increased by TGF-β. This effect was attributed to the inhibition of inflammatory signaling pathways such as IL-6 signaling. Based on these effects, a clinical trial is currently underway to investigate the association of MT-7117 with SSc patients.

#### 4.3.3. AP1189 (Resomelagon)

A series of pyrrole aminoguanidine derivatives were investigated as ligands in the melanocortin receptors by Lundstedt et al. In order to assess the activity of melanocortin receptors, researchers measured the binding affinities (Ki) of pyrrole aminoguanidine derivatives for MC1R, MC3R, MC4R, and MC5R. While most of the compounds did not display significant activity or selectivity, the phenylpyrrole allylidene derivatives exhibited enhanced binding affinities towards MC1R and therefore, meaningful selectivities. The phenyl group was found to be particularly important for binding to MC1R. AP1189 ([1-(4-chlorophenyl)-1H-pyrrol-2-yl-allylideneamino]guanidinium acetate) was identified as an agonist for melanocortin receptors MC1R and MC3R. (Figure 11) [119,124]. 

AP1189 has a unique mechanism that distinguishes it from other compounds. It does not stimulate cAMP production and is not involved in melanogenesis. Instead, it induces phosphorylation of ERK1/2, resulting in a decrease in cytokine levels in macrophages. This may help resolve acute inflammation and potentially alleviate arthritis in mice [125]. Based on these findings, AP1189 is currently being evaluated in clinical trials for the treatment of rheumatoid arthritis and idiopathic membranous nephropathy [67].

#### 4.3.4. CD08108

Selective novel MC1R agonists were discovered by the research group in GALDERMA research and development in 2013 [120]. Imidazolyl-linked azetidine derivatives were synthesized and evaluated for activity for human melanocortin receptors. Among them, CD08108 exhibited a significant hMC1R agonistic activity with EC50 of 70 nM and showed the selectivity of MC1R over MC4R up to 64 times (Figure 12). In 2015, Boiteau et al. reported an efficient synthetic route for CD08108 in a kilogram scale [126]. CD08108 has been implicated in the process of melanogenesis, indicating its potential as a therapeutic target for the treatment of skin conditions such as hypodermic or photosensitive diseases.

#### 4.3.5. JNJ-10229570

2,3-Diphenyl-5-anilino [1,2,4]thiadiazole has been confirmed to act as an agonist for MC4R, thereby inhibiting food intake and affecting eating behavior in mice. JNJ-10229570 is a compound with 2,3-(2-methoxyphenyl) at 2,3-Diaryl and reported to function as an antagonist for MC1R and MC5R, which are melanocortin receptors expressed in sebocytes (Figure 13). Melanocortin receptors in sebocytes are known to play a crucial role in producing sebaceous lipids. Studies on the relationship between MC5R activation and sebocytes indicate that the activation of MC5R increases the production of cAMP, inducing the production of sebaceous lipids [127,128,129]. Sebaceous lipids are essential for skin integrity and inflammatory processes. However, excessive sebum production is shown to be a significant contributor to the pathophysiology of Acne vulgaris [130,131,132]. Eisinger et al. demonstrated that JNJ-10229570 acts as an antagonist for MC1R and MC5R present in human sebaceous cells, inhibiting sebaceous secretion and reducing sebaceous gland size [121]. The IC_50_ measurements for MC1R and MC5R, using 125I-NDP-α-MSH, were found to be 270 ± 120 and 200 ± 50 nM (Mean ± S.D.), respectively, indicating that JNJ-10229570 binds to both receptors. Furthermore, the reduction in the amount of cAMP, elevated by NDP-MSH after the sebocyte was exposed to 0.6 nM of JNJ-10229570, suggests that this compound acts as an antagonist for MC1R and MC5R. In addition, human skin transplanted into SCID mice and treated with JNJ-10229570 showed a decrease in squalene, wax esters, and triglyceride, suggesting new information about the effect of JNJ-10229570 on surface lipid secretion in vivo.

## 5. Conclusions

The discovery of the melanocortin 1 receptor several decades ago has led to extensive research into its potential as a therapeutic target. Studies have revealed that the MC1R signaling pathway is crucial in treating melanoma, as it affects various physiological changes in melanocytes. However, recent research has also shown a growing interest in investigating the molecular mechanisms responsible for the non-pigmentary effects of MC1R, particularly in the treatment of inflammatory diseases. Ongoing investigations suggest that drugs targeting MC1R may hold promise in treating patients with high-need conditions such as systemic sclerosis, neuroinflammation, rheumatoid arthritis, fibrosis, and others. With numerous innovative molecules and clinical trials currently underway, it is possible that these drugs may soon improve the quality of life for those with chronic inflammatory conditions. Furthermore, the development of MC1R ligands using multiple chemistry approaches and knowledge of the receptor’s crystal structure may lead to the creation of additional drugs to treat new conditions in the future.

## Figures and Tables

**Figure 1 ijms-24-12152-f001:**
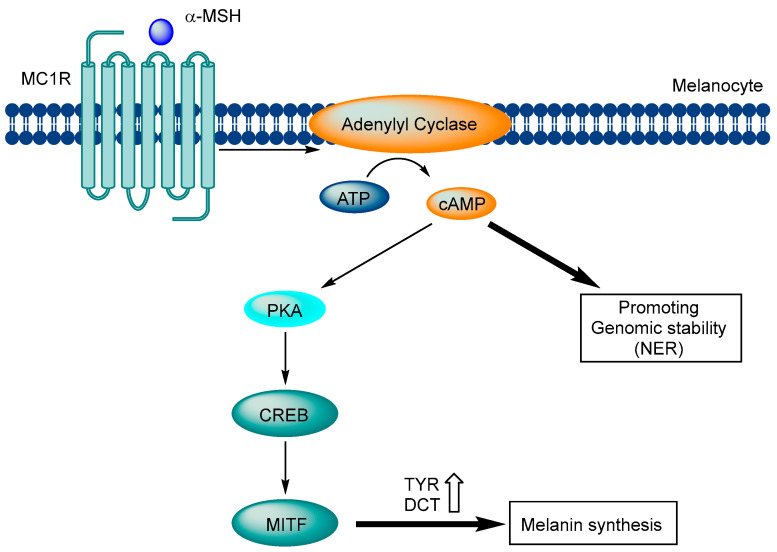
Signaling pathway of MC1R in melanocytes.

**Figure 2 ijms-24-12152-f002:**
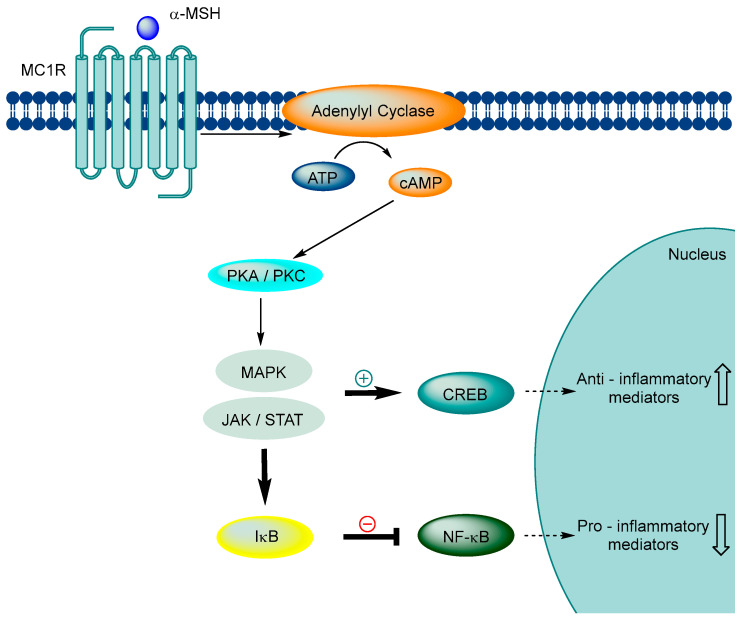
The role of MC1R signaling in anti-inflammatory effects.

**Figure 3 ijms-24-12152-f003:**
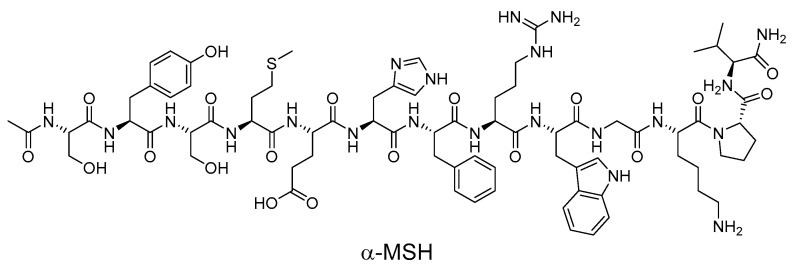
Structure of α-MSH.

**Figure 4 ijms-24-12152-f004:**
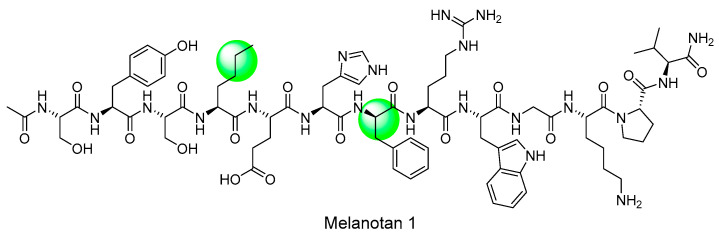
Structure of melanotan I.

**Figure 5 ijms-24-12152-f005:**
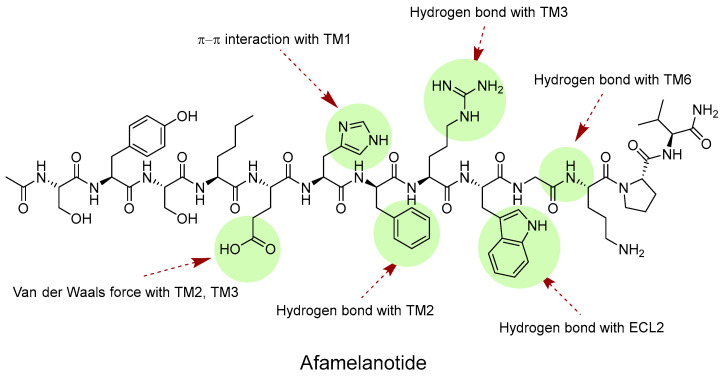
Interactions of afamelanotide (melanotan I) with MC1R domains.

**Figure 6 ijms-24-12152-f006:**
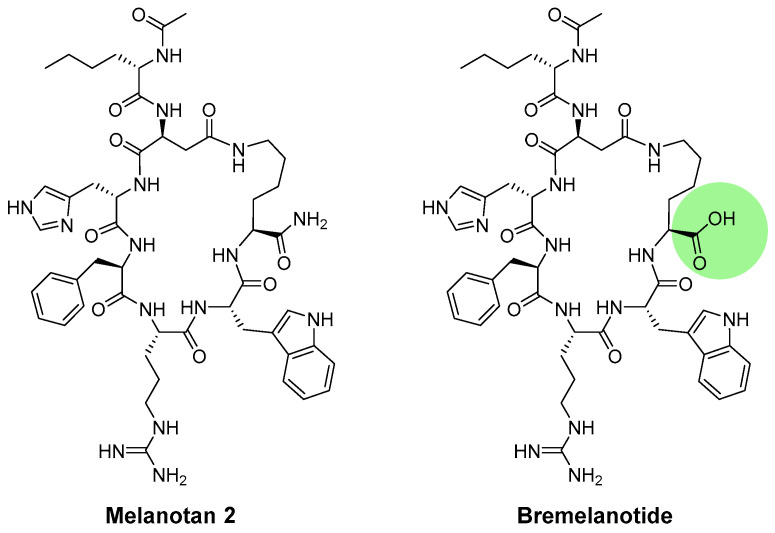
Structure of melanotan II and bremelanotide.

**Figure 7 ijms-24-12152-f007:**
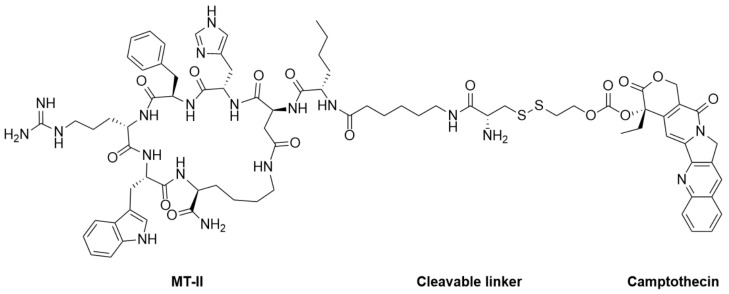
Structure of Camptothecin-MT-II conjugate.

**Figure 8 ijms-24-12152-f008:**
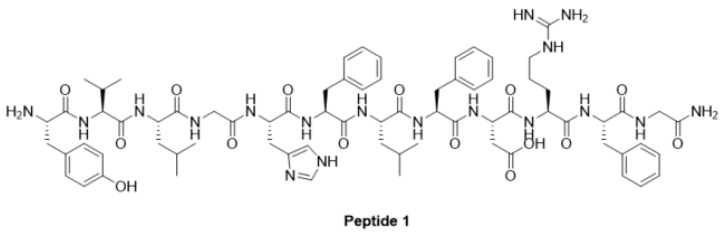
Structure of Peptide 1.

**Figure 9 ijms-24-12152-f009:**
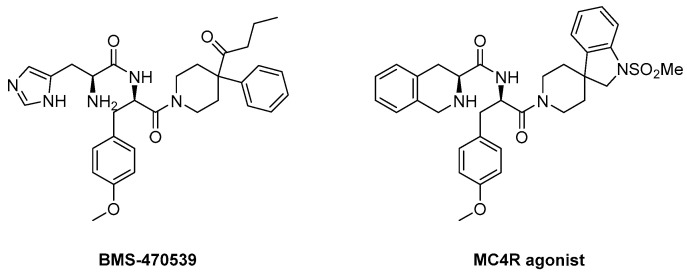
Structure of BMS-470539 and MC4R agonist.

**Figure 10 ijms-24-12152-f010:**
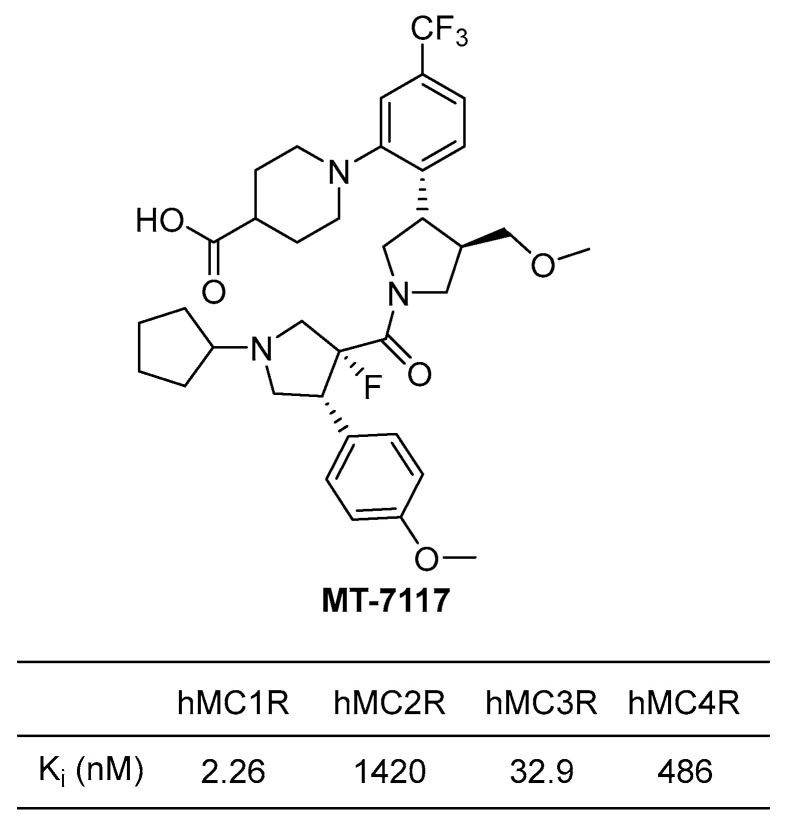
Binding affinities (*K*i) of MT-7117 (dersimelagon).

**Figure 11 ijms-24-12152-f011:**
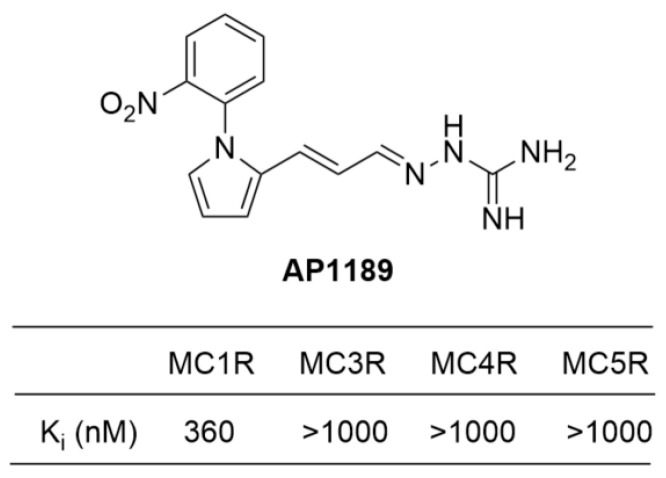
Structure and binding affinity of AP1189.

**Figure 12 ijms-24-12152-f012:**
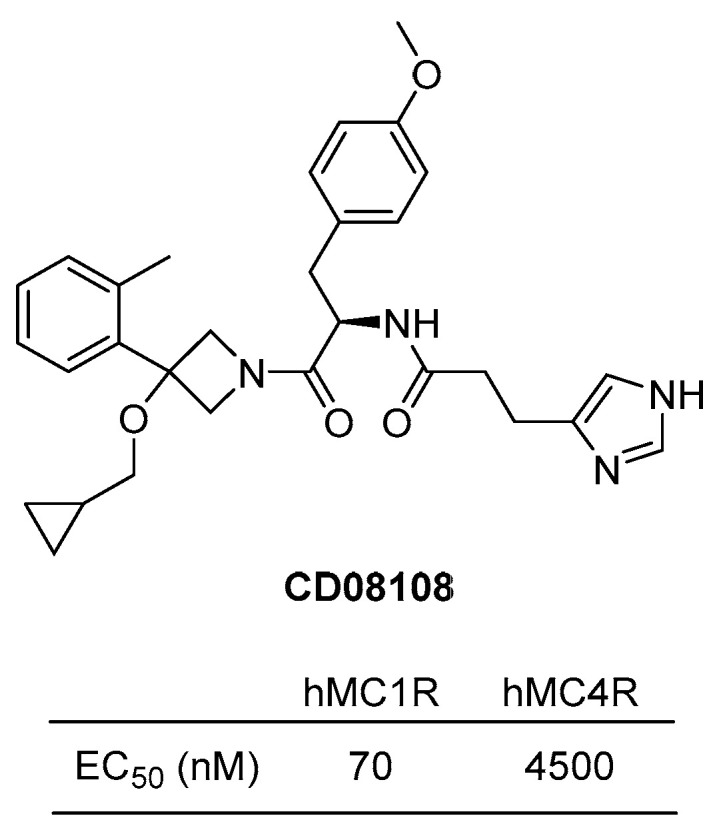
Structure of CD08108.

**Figure 13 ijms-24-12152-f013:**
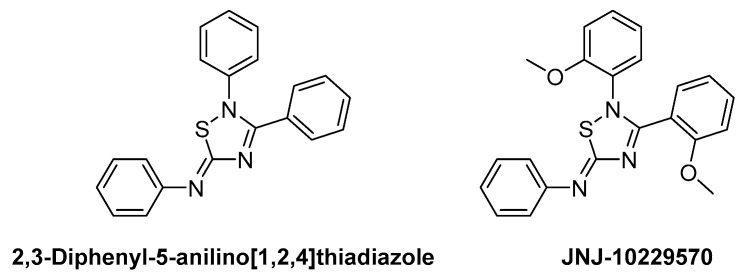
Structures of 2,3-Diphenyl-5-anilino [1,2,4]thiadiazole and JNJ-10229570.

**Table 1 ijms-24-12152-t001:**
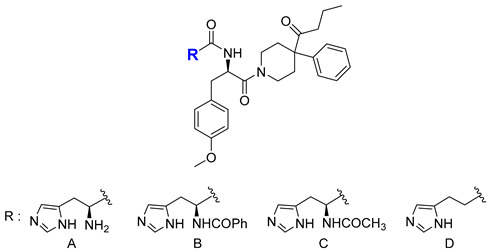
SAR of side chain of BMS-470539.

	MC1R	MC3R	MC4R	MC5R
Compound	EC_50_(nM)	EC_50_(nM)	EC_50_(nM)	EC_50_(nM)
A	28 ± 12	NA	2600 ± 200	4400 ± 1300
B	0.19 ± 0.02	250 ± 170	2.9 ± 0.04	2100 ± 400
C	0.35 ± 0.07	2200 ± 1000	23 ± 12	3900 ± 900
D	2.5 ± 1.3	7000 ± 1700	840 ± 150	9400 ± 200

## Data Availability

Not applicable.

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
