# Peer review of "Melanocortin 1 Receptor (MC1R): Pharmacological and Therapeutic Aspects"

_ijms, 2023, doi:10.3390/ijms241512152_

Round 1

Reviewer 1 Report

Comments and Suggestions for Authors

This manuscript reviews the studies about to updated the recent knowledge about Melanocortin 1 Receptor (MC1R) and MC1R-targeted treatment of various diseases. 

It is a nice review. The manuscript is technically correct. The introduction outlines the problem and the research objective. The manuscript is well organized and clearly presents the main results in the related field. In particular, the topic is very important for practical applications  (for example in cosmetology for the treatment of hyperpigmentory disorders and in medicine). In my opinion, the research responds to the need to systematise the scattered knowledge on melanocortin 1 receptor (MC1R) and its importance in the medical therapy of skin conditions. I think the theme is original. It is important and relevant in the fields of health sciences, chemical engineering and the field of pharmaceutical sciences, cosmetology and  in medical science. 

The article does meet the editorial standards of the journal. English language and style are fine. The abstract does give a full clarity of what is being article. The introduction is correctly spelled. The tables  and figures contain all the necessary information, are appropriately captioned and clear. Units and abbreviations are explicit. The conclusions are consistent with the evidence and arguments presented. The reference materials are well-selected and up to date, although I am still missing items from 2019-2022. To sum up, the review is suggested to be published in IJMS.  In general, the topic is interesting, the approach is suitable for a this journal.

Author Response

We attached the response to the reviewer's comments.

Reviewer 2 Report

Comments and Suggestions for Authors

The reviewer read the manuscript entitled "Melanocortin 1 Receptor (MC1R): Potentials as Therapeutic Targets", by Yoon Woo Mun et al., submitted to Section "Molecular Pharmacology", Special Issue "Advances in Molecular Activity of Potential Drugs 3.0". At the first look, the manuscript is aiming to provide a comprehensive review of the topic of the Melanocortin 1 Receptor (MC1R), with a focus on its signaling pathways and related diseases. However, it is neither comprehensive neither novel.

Lot of issues need to be corrected before the final decision about publication. 

A) Title: manuscript is entitled: Melanocortin 1 Receptor (MC1R): Potentials as Therapeutic Targets. It obliges to put more insight into pharmacological and terapeutic aspects.

B) The authors cited not very recent scientific papers. Furthermore, they also cited a RETRACTED ARTICLE: REF. 36. Jarrett, S.G.; Horrell, E.M.W.; Christian, P.A.; Vanover, J.C.; Boulanger, M.C.; Zou, Y.; D'Orazio, J.A. PKA-mediated phosphorylation of ATR promotes recruitment of XPA to UV-induced DNA damage. Mol Cell 2014, 54, 999-1011, doi:10.1016/j.molcel.2014.05.030.

C) The major problem is that the Authors very often cite review articles, not the original research publications. 

Bibliography which refers to the review not to the orginal papers:

1. Cawley, N.X.; Li, Z.; Loh, Y.P. 60 YEARS OF POMC: Biosynthesis, trafficking, and secretion of pro-opiomelanocortin-derived peptides. J Mol Endocrinol 2016, 56, T77-97, doi:10.1530/JME-15-0323.

2. Dores, R.M.; Liang, L.; Davis, P.; Thomas, A.L.; Petko, B. 60 YEARS OF POMC: Melanocortin receptors: evolution of ligand selectivity for melanocortin peptides. J Mol Endocrinol 2016, 56, T119-133, doi:10.1530/JME-15-0292.

3. Gallo-Payet, N. 60 YEARS OF POMC: Adrenal and extra-adrenal functions of ACTH. J Mol Endocrinol 2016, 56, T135-156, doi:10.1530/JME-15-0257.

4. Lipton, J.M.; Ceriani, G.; Macaluso, A.; McCoy, D.; Carnes, K.; Biltz, J.; Catania, A. Antiinflammatory effects of the neuropeptide alpha-MSH in acute, chronic, and systemic inflammation. Ann N Y Acad Sci 1994, 741, 137-148, doi:10.1111/j.1749-6632.1994.tb39654.x.

5. Catania, A.; Cutuli, M.; Garofalo, L.; Carlin, A.; Airaghi, L.; Barcellini, W.; Lipton, J.M. The neuropeptide alpha-MSH in host defense. Ann N Y Acad Sci 2000, 917, 227-231, doi:10.1111/j.1749-6632.2000.tb05387.x.

11. Yang, Y. Structure, function and regulation of the melanocortin receptors. Eur J Pharmacol 2011, 660, 125-130, doi:10.1016/j.ejphar.2010.12.020.

12. Wolf Horrell, E.M.; Boulanger, M.C.; D'Orazio, J.A. Melanocortin 1 Receptor: Structure, Function, and Regulation. Front Genet 2016, 7, 95, doi:10.3389/fgene.2016.00095.

28. Garcia-Borron, J.C.; Sanchez-Laorden, B.L.; Jimenez-Cervantes, C. Melanocortin-1 receptor structure and functional regulation. Pigment Cell Res 2005, 18, 393-410, doi:10.1111/j.1600-0749.2005.00278.x.

29. Pitcher, J.A.; Freedman, N.J.; Lefkowitz, R.J. G protein-coupled receptor kinases. Annu Rev Biochem 1998, 67, 653-692, doi:10.1146/annurev.biochem.67.1.653.

30. Luttrell, L.M.; Lefkowitz, R.J. The role of beta-arrestins in the termination and transduction of G-protein-coupled receptor signals. J Cell Sci 2002, 115, 455-465, doi:10.1242/jcs.115.3.455.

31. Strader, C.D.; Fong, T.M.; Tota, M.R.; Underwood, D.; Dixon, R.A. Structure and function of G protein-coupled receptors. Annu Rev Biochem 1994, 63, 101-132, doi:10.1146/annurev.bi.63.070194.000533.

32. Qanbar, R.; Bouvier, M. Role of palmitoylation/depalmitoylation reactions in G-protein-coupled receptor function. Pharmacol Ther 2003, 97, 1-33, doi:10.1016/s0163-7258(02)00300-5.

34. Holst, B.; Schwartz, T.W. Molecular mechanism of agonism and inverse agonism in the melanocortin receptors: Zn(2+) as a structural and functional probe. Ann N Y Acad Sci 2003, 994, 1-11, doi:10.1111/j.1749-6632.2003.tb03156.x.

40. Abdel-Malek, Z.A.; Swope, V.B.; Starner, R.J.; Koikov, L.; Cassidy, P.; Leachman, S. Melanocortins and the melanocortin 1 receptor, moving translationally towards melanoma prevention. Arch Biochem Biophys 2014,563,4-12, doi:10.1016/j.abb.2014.07.002.

42. Dorsam, R.T.; Gutkind, J.S. G-protein-coupled receptors and cancer. Nat Rev Cancer 2007, 7, 79-94, doi:10.1038/nrc2069.

43. Neves, S.R.; Ram, P.T.; Iyengar, R. G protein pathways. Science 2002, 296, 1636-1639, doi:10.1126/science.1071550.

44. Herraiz, C.; Garcia-Borron, J.C.; Jimenez-Cervantes, C.; Olivares, C. MC1R signaling. Intracellular partners and pathophysiological implications. Biochim Biophys Acta Mol Basis Dis 2017, 1863, 2448-2461, doi:10.1016/j.bbadis.2017.02.027.

46. Tanaka, T.; Narazaki, M.; Kishimoto, T. IL-6 in inflammation, immunity, and disease. Cold Spring Harb Perspect Biol 2014, 6, a016295, doi:10.1101/cshperspect.a016295.

58. Gerlo, S.; Kooijman, R.; Beck, I.M.; Kolmus, K.; Spooren, A.; Haegeman, G. Cyclic AMP: a selective modulator of NF-kappaB action. Cell Mol Life Sci 2011, 68, 3823-3841, doi:10.1007/s00018-011-0757-8.

60. Wang, W.; Guo, D.Y.; Lin, Y.J.; Tao, Y.X. Melanocortin Regulation of Inflammation. Front Endocrinol (Lausanne) 2019, 10, 683, doi:10.3389/fendo.2019.00683.

61. Lo, J.A.; Fisher, D.E. The melanoma revolution: from UV carcinogenesis to a new era in therapeutics. Science 2014, 346, 945-949, doi:10.1126/science.1253735.

62. Rosenkranz, A.A.; Slastnikova, T.A.; Durymanov, M.O.; Sobolev, A.S. Malignant melanoma and melanocortin 1 receptor. Biochemistry (Mosc) 2013, 78, 1228-1237, doi:10.1134/S0006297913110035.

65. Eggermont, A.M. Advances in systemic treatment of melanoma. Ann Oncol 2010, 21 Suppl 7, vii339-344, doi:10.1093/annonc/mdq364.

67. Montero-Melendez, T.; Boesen, T.; Jonassen, T.E.N. Translational advances of melanocortin drugs: Integrating biology, chemistry and genetics. Semin Immunol 2022, 59, 101603, doi:10.1016/j.smim.2022.101603.

70. Denton, C.P.; Ong, V.H. Targeted therapies for systemic sclerosis. Nat Rev Rheumatol 2013, 9, 451-464, 614 doi:10.1038/nrrheum.2013.46.

71. Allanore, Y.; Simms, R.; Distler, O.; Trojanowska, M.; Pope, J.; Denton, C.P.; Varga, J. Systemic sclerosis. Nat Rev Dis Primers 2015, 1, 15002, doi:10.1038/nrdp.2015.2.

75. Dixon, B.J.; Reis, C.; Ho, W.M.; Tang, J.; Zhang, J.H. Neuroprotective Strategies after Neonatal Hypoxic Ischemic Encephalopathy. Int J Mol Sci 2015, 16, 22368-22401, doi:10.3390/ijms160922368.

81. Catania, A.; Gatti, S.; Colombo, G.; Lipton, J.M. Targeting melanocortin receptors as a novel strategy to control inflammation. Pharmacol Rev 2004, 56, 1-29, doi:10.1124/pr.56.1.1.

84. Moscowitz, A.E.; Asif, H.; Lindenmaier, L.B.; Calzadilla, A.; Zhang, C.; Mirsaeidi, M. The Importance of Melanocortin Receptors and Their Agonists in Pulmonary Disease. Front Med (Lausanne) 2019, 6, 145, doi:10.3389/fmed.2019.00145.

107. Lane, A.M.; McKay, J.T.; Bonkovsky, H.L. Advances in the management of erythropoietic protoporphyria - role of afamelanotide. Appl Clin Genet 2016, 9, 179-189, doi:10.2147/TACG.S122030.

108. Hadley, M.E.; Hruby, V.J.; Blanchard, J.; Dorr, R.T.; Levine, N.; Dawson, B.V.; al-Obeidi, F.; Sawyer, T.K. Discovery and development of novel melanogenic drugs. Melanotan-I and -II. Pharm Biotechnol 1998, 11, 575-595, doi:10.1007/0-306-47384-4_25.

113. Shadiack, A.M.; Sharma, S.D.; Earle, D.C.; Spana, C.; Hallam, T.J. Melanocortins in the treatment of male and female sexual dysfunction. Curr Top Med Chem 2007, 7, 1137-1144, doi:10.2174/156802607780906681.

116. Minder, E.I. Afamelanotide, an agonistic analog of alpha-melanocyte-stimulating hormone, in dermal phototoxicity of erythropoietic protoporphyria. Expert Opin Investig Drugs 2010, 19, 1591-1602, doi:10.1517/13543784.2010.535515.

131. Zouboulis, C.C.; Schagen, S.; Alestas, T. The sebocyte culture: a model to study the pathophysiology of the sebaceous gland in sebostasis, seborrhoea and acne. Arch Dermatol Res 2008, 300, 397-413, doi:10.1007/s00403-008-0879-5.

132. Thiboutot, D.; Gollnick, H.; Bettoli, V.; Dreno, B.; Kang, S.; Leyden, J.J.; Shalita, A.R.; Lozada, V.T.; Berson, D.; Finlay, A.; et al. New insights into the management of acne: an update from the Global Alliance to Improve Outcomes in Acne group. J Am Acad Dermatol 2009, 60, S1-50, doi:10.1016/j.jaad.2009.01.019.

The manuscript lacks novelty. Very similar articles have been already published and the reviewer does not see what this manuscript adds to existing literature. The decision to publish or not depends on the policy of the Editor.

Author Response

We attached the file "Response to the reviewer's comments".

Reviewer 3 Report

Comments and Suggestions for Authors

The authors well-described the review article related to MC1R receptor.

I could not find any downside in this manuscript. 

So, I think this manuscript is acceptable in IJMS.

Author Response

We attachen the file, "Response to the reviewer's comments".

Reviewer 4 Report

Comments and Suggestions for Authors

I find the topic of the review article titled "Melanocortin 1 Receptor (MC1R): Potentials as Therapeutic Targets" very interesting and valuable. The authors presented fundamental information on melanocortins, as well as the structure and function of MC1R. They also described the diseases associated with MC1R and modulators of this receptor. However, I believe it would be worthwhile to additionally describe the potential involvement of drugs targeting the MC1R receptor in the treatment of pigmentation disorders (I also suggest characterizing such disorders). It would be valuable to summarize certain limitations of drugs targeting MC1R (non-specificity of action on signaling pathways, possible side effects, pharmacokinetic properties - the need for penetration into the basal layer of the epidermis, and application possibilities).

Comments on the Quality of English Language

Minor editing of English language required

Author Response

We attached the file, "Response to the reviewer's comments".

Round 2

Reviewer 2 Report

Comments and Suggestions for Authors

The reviewer remains of his opinion. He leaves it to the editor to decide whether or not to accept the manuscript.